# MMCFormer: Missing Modality Compensation Transformer for Brain Tumor Segmentation

**Sanaz Karimijafarbigloo** [1,2]                      SANAZ.KARIMIJAFARBLOO@UR.DE

**Reza Azad** [2]                                   REZA.AZAD@LFB.RWTH-AACHEN.DE

**Amirhossein Kazerouni** [3]                              AMIRHOSSEIN477@GMAIL.COM

**Saeed Ebadollahi** [3]                                 S_EBADOLLAHI@IUST.AC.IR

**Dorit Merhof** [1,4]                                  DORIT.MERHOF@UR.DE

[1] *Faculty of Informatics and Data Science, University of Regensburg, Regensburg, Germany*

[2] *Institute of Imaging and Computer Vision, RWTH Aachen University, Aachen, Germany*

[3] *School of Electrical Engineering, Iran University of Science and Technology, Tehran, Iran*

[4] *Fraunhofer Institute for Digital Medicine MEVIS, Bremen, Germany*

**Editors:** Accepted for publication at MIDL 2023

## Abstract

Human brain tumours and more specifically gliomas are amongst the most life-threatening cancers which usually arise from abnormal growth of the glial stem cells. In practice, Magnetic Resonance Imaging (MRI) modalities, which offer different contrasts to elucidate tissue properties, provide comprehensive information regarding the brain's structure and also potential clues for detecting tumors. Hence, multi-modal MRI is commonly utilized for the diagnosis of brain tumors. However, since the set of acquired modalities may vary between clinical sites, brain tumor studies may miss one or two MRI modalities. To address missing information in an end-to-end manner, we propose MMCFormer, a novel missing modality compensation network. Our strategy builds upon 3D efficient transformer blocks and uses a co-training strategy to effectively train a missing modality network. To ensure feature consistency in a multi-scale fashion, MMCFormer utilizes global contextual agreement modules in each scale of the encoders. Furthermore, to transfer modality-specific representations, we propose to incorporate auxiliary tokens in the bottleneck stage to model interaction between full and missing-modality paths. On top of that, we include feature consistency losses to reduce the domain gap in network prediction and increase the prediction reliability for the missing modality path. Extensive experiments on the BraTS 2018 dataset demonstrate the benefits of our approach compared to competing approaches. The implementation code is publicly available at GitHub.

**Keywords:** Transformer, Missing Modality, Segmentation, MRI, Medical.

## 1. Introduction

Magnetic Resonance Imaging (MRI) is a prevalent non-invasive imaging utility commonly used in medical diagnosis and treatment that can provide a 3D representation of human organs, tissues, and the skeletal system. However, interpretation of the data by a clinical expert (e.g. localizing and segmenting the brain tumor) is typically time-consuming and expensive. In addition, due to inter- and intra-rater variabilities, clinical expert annotations are always accompanied by uncertainties. Therefore, multi-modal MRI imaging (e.g.

T1, T1c, T2, and Flair modalities) can be beneficial since it aggregates complementary modality-specific information and allows more accurate pathology assessment. However, acquiring all desirable imaging modalities sometimes is not feasible due to practical constraints, such as lengthy scan time and image corruption. Hence, a missing modality issue may raise uncertainty for tumour localization and segmentation, since each modality contributes unique and comprehensive information (Azad et al., 2022b; Zhao et al., 2022).

Several methods have been presented to solve the problem of automatic segmentation. Over the past decade, Convolutional Neural Networks (CNNs) have been instrumental in the development of medical image segmentation, such as UNet (Ronneberger et al., 2015), and its variants (Valanarasu et al., 2020; Zhou et al., 2018; Azad et al., 2022a), which have been widely used due to their symmetric architecture and potentiality to capture contextual semantic information while preserving multi-scale features with skip connections. Recently, the introduction of the Vision Transformer (ViT) has substantially improved the performance of all computer vision tasks by including an attention mechanism that effectively captures global information (Dosovitskiy et al., 2020). Since then, Transformers have been diligently deployed in medical image segmentation to compensate for CNNs' inability to capture sufficient contextual information, yielding poor performance in boundary areas. Motivated by the U-shaped architecture of UNet, Swin-UNet (Cao et al., 2021), and DS-TransUNet (Lin et al., 2022) propose pure transformer models based on Swin Transformer for image segmentation. Similarly, MISSFormer (Huang et al., 2021) employs a hierarchical symmetric encoder-decoder Transformer coupled with the enhanced Transformer block to boost the feature representations. Aside from fully-Transformer models, various approaches have been presented that employ both CNNs and Transformers together to acquire low-level and high-level features (Heidari et al., 2022; Chen et al., 2021). However, these models require the presence of all modalities to provide their excellent results. Otherwise, in case of a missing modality, their performance would decrease.

Several methods have been proposed for dealing with missing data in medical imaging. Early approaches aim at synthesizing the missing modality during the training process (Azad et al., 2022b). However, reconstructing the missing modality usually requires designing a modality-specific network and limits the performance of these strategies for end-to-end training. Therefore, more recent methods adopt a more efficient strategy, building a uniform model for all possible missing modalities by learning full modality joint representations. HEMIS (Havaei et al., 2016) proposes to encode each modality into an embedding space using separate convolutional layers. Afterward, the embedded features' first and second-order moments (mean and variance) are calculated to establish a common representation space between modalities. During inference, any combination of inputs can be given to construct the segmentation map. The same strategy is followed by PIMMS (Varsavsky et al., 2018), HVED (Dorent et al., 2019), and URN (Lau et al., 2019) in creating a common latent subspace and seeking to retrieve the missing information using the constructed latent representation. However, such approaches are unable to provide adequate information using only mean and variance and yield poor performance when more than one modality is absent.

In recent years, Generative Adversarial Networks (GANs) have been established for generating synthetic data (Qin et al., 2022; Chang et al., 2020; Cao et al., 2020; Sharma and Hamarneh, 2019). Nevertheless, employing GAN-based models to tackle the missing modality problem raises several difficulties. In addition to possibly generating undesirable

imputation noise when synthesizing the missing modality, they can also show non-converging characteristics, require extensive and careful training, and the generator is likely to become unstable during training.

Other approaches, such as the networks described by (Azad et al., 2022c; Liu et al., 2022; Wang et al., 2021), take advantage of the knowledge distillation strategy. They offer a co-training strategy for training the "complete modality" and "missing modality" models simultaneously in order to complement each other's feature representations and restore missing knowledge of missing modalities. However, their models rely on the CNN structure and are therefore unable to effectively capture structural and shape information (e.g., learning brain structure and detecting abnormal regions) from the "full modality" path, resulting in less accurate prediction when facing several missing modalities. Besides that, these methods only perform feature recalibration between the "full modality" and the "missing modality" paths and lack to imitate the modality-specific information and contextual consistency. In this paper, we argue that contextual consistency is crucial for the segmentation task and seek to model such consistency with the Transformer model. Furthermore, we propose to modify the bottleneck representation by including additional tokens to learn modality-specific representation and compensate for the missing information. We create our architecture in a pure Transformer fashion with efficient self-attention blocks and perform the co-training strategy in an end-to-end manner.

## 2. Proposed Method

The overall structure of the proposed MMCFormer is depicted in Figure 1. MMCFormer is a novel pure Transformer-based approach that adopts the co-training strategy for the MRI missing modality task, where one network is assigned to train the complete modality while the other trains with the missing modality. Co-training is a multi-view algorithm that predicts every data point from two independent views (Rahate et al., 2022). These two views are trained independently and complement one another since both provide unique, complementary information. On the other hand, MRI modalities have shown that they can provide comprehensive and distinguishable features of brain structure that enable physicians to have more accurate anatomical and functional examinations. To this end, taking an input 3D modality image $X_i \in R^{H \times W \times D \times C}$ with spatial dimensions $H \times W \times D$ and $C$ channels (i.e. modalities) and the set of modalities M = {T1, T1w, T2 and Flair}, we can describe the full-modality data as $X_f$, which for all subjects it contains all the modalities in M, and the missing modality as $X_m$, which for each subject it may contain one or more modalities from M. We define two parallel 3D efficient Transformers U-Net models for our co-training strategy, where each network is trained separately and simultaneously. One is trained on the complete data $(X_f)$ to obtain comprehensive information about the full modality, and the second one is trained on the incomplete data $(X_m)$ to derive the missing modalities details. The co-training strategy allows the "full modality" network to distill its knowledge into the "missing modality" one, and compensate for the lack of modality-specific information. In the next subsections, we first present our network structure and then present our knowledge distillation mechanisms.

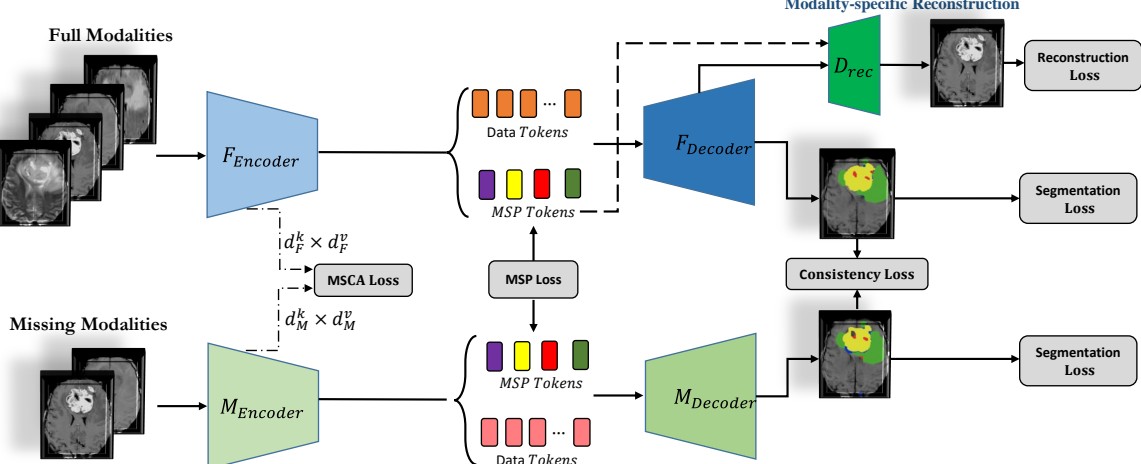

Figure 1: Overview of the proposed MMCFormer. MMCFormer deploys three feature-matching mechanisms to reduce the domain gap and to ensure knowledge distillation from the "full modality" path into a "missing modality" network.

## 2.1. Network Architecture

As stated earlier, our network builds upon an efficient Transformer design to reduce the computational burden of 3D data processing. More precisely, for an input image $X^{H \times W \times D \times C}$, we first perform the tokenization process by extracting $n$ non-overlapping 3D patches and applying the parametric mapping function to embed each token into a $d$ dimensional space. Next, following the general idea of the self-attention mechanism, we create the $Q$, $K$, and $V$ matrices using linear projections. However, instead of applying the basic self-attention module, which suffers from a quadratic computational complexity, we follow the efficient self-attention strategy (Shen et al., 2021)

$$E(Q, K, V) = \rho_q(Q) \left( \rho_k(k)^T V \right), \tag{1}$$

where $\rho$ shows the softmax operation. It was shown that (Shen et al., 2021) the efficient self-attention module can approximate the standard self-attention operation but with much lower computational complexity, $O(d^2 n)$ vs $O(n^2)$ (Shen et al., 2021). Where $d$ and $n$ indicate the embedding dimension and the number of tokens, respectively. For our U-Net structure, we follow a symmetrical design and include three efficient Transformer blocks in each encoder/decoder block. We use patch merging and patch expanding layers in the encoding and decoding paths, respectively. Finally, we include the skip-connection path in each scale of the network to capture multi-scale representations. The same structure is used in both the "full modality" and the "missing modality" network. For training, two main and three auxiliary loss functions are applied to reconstruct the segmentation map and ensure the knowledge distillation between the full modality and the missing modality paths. Our first primary loss function is the Dice loss, which computes the loss between the

ground truth $Y$ and the predicted mask $Y'$ to learn the segmentation map:

$$\mathcal{L}_{\text{seg}} = \alpha \mathcal{L}_{\text{dice}} \left( Y'_f, Y \right) + \beta \mathcal{L}_{\text{dice}} \left( Y'_m, Y \right) \tag{2}$$

In the training step, we set $\alpha$ to 0.6, and $\beta$ to 0.4. In the second step, we follow (Azad et al., 2022c) and design a consistency loss function to reduce the domain gap between the "full modality" and the missing one, aiming to reduce the distance between the two distributions and increase the confidence of the "missing modality" model. Thus, our approach will improve the boundary area estimation in cases where the uncertainty (or entropy) of pixels is high or when the probability of a pixel belonging to a class is low. Equation 3 illustrates our consistency loss function:

$$\mathcal{L}_{Consistency} \left( S_f, S_m \right) = \sum_{i=1}^{c} \left| S_f^i - S_m^i \right| \tag{3}$$

where, $S^i$ indicates the soft prediction map in the $i_{th}$ channel of the network in each path. Next, we design and employ two knowledge distillation modules, namely Multi-Scale Contextual Agreement (MSCA) and Modality-Specific (MSP) tokens, to ensure the effective distillation of the semantic and contextual information associated with the full-modality model into the missing one, as outlined in the next subsection.

### 2.2. Knowledge Distillation Modules

**Multi-Scale Contextual Agreement (MSCA) Module**: Considering a rich representation of the full-modality path, we aim to impose a hierarchical knowledge distillation module to the missing-modality path to adaptively recalibrate its feature representation and minimize the domain gap. To this end, we propose the MSCA module, which seeks to minimize the context distribution between the missing and full-modality paths in each scale of the network. In our efficient self-attention mechanism, we compute the global context (GC) using the query and value vectors as $GC = (\rho_k(k)^T V) \in R^{d \times d}$. To effectively align the correlation matrix calculated in both paths ($GC_f$ and $GC_m$), we strive to minimize the distance as follows:

$$\mathcal{L}_{\text{MSCA}} \left( \mathbf{GC}_f, \mathbf{GC}_m \right) = 1 - \frac{\text{tr} \left\{ \mathbf{GC}_f \mathbf{GC}_m \right\}}{\| \mathbf{GC}_f \| \, \| \mathbf{GC}_m \|} \quad \in [0, 1] \tag{4}$$

The $\mathcal{L}_{\text{MSCA}}$ becomes zero when the correlation matrices are equal and one if they have maximum difference. Using the inner product, the above equation can be considered as cosine dissimilarity: $\mathcal{L}_{\text{MSCA}}(\mathbf{GC_f}, \mathbf{GC_m}) = 1 - \cos \left( \mathbf{GC}_f, \mathbf{GC}_m \right)$. The design of $\mathcal{L}_{MSCA}$ is well suited to align the correlation of the "missing modality" path with the "full-modality" one.

**Modality Specific (MSP) Module**: In the MSCA module, we aimed to align the feature distribution between the two networks. However, such alignment might suppress the discriminative modality-specific (MS) features that existed in the full-modality paths. Therefore, we design the MSP module to preserve discriminative features in the full path and transfer this information to the missing-modality path. For this purpose, we define four modality-specific tokens in the network bottleneck, where each token aims to preserve distinctive features of each input modality. We choose bottleneck since it contains highly

condensed low-dimensional information of label maps, hence, we can exploit its useful and distinctive features to distill the modality-specific knowledge to the "missing modality" network. Besides that the bottleneck layer summarizes the rich feature representation of the encoder into a lower-dimensional and more compact representation, making it easier to match the modalities and maintain consistency between the two networks. Thus, MSP loss is calculated using the output of full and missing tokens from the bottleneck, culminating in transferring modality-specific distinguishing features from the complete model to the incomplete one. The MSP loss is defined as:

$$\mathcal{L}_{MSP}\left(MSP_f, MSP_m\right) = \sum_{i=1}^{M} |MSP_f^i - MSP_m^i| \tag{5}$$

This soft-aligned feature representation helps the missing-modality path to reconstruct the discriminative features which are not feasible through its incomplete input modalities. We finally devise a reconstruction loss in the full-modality path to ensure that each token learns modality-specific information. To do so, we employ an auxiliary reconstruction decoder head ($D_{rec}$) alongside the segmentation head to perform the reconstruction task. In each step, we randomly take one of the tokens alongside the resultant feature maps from the previous decoder layer to input into the reconstruction decoder, seeking to reconstruct the modality assigned to the class token. By doing so, we can guarantee that each token learns distinctive modality information. For the reconstruction loss, we use the combination of MS-SSIM and L1 distance (Zhao et al., 2016) to better reconstruct the high-frequency information alongside the image content, which are crucial for the MRI modalities:

$$\mathcal{L}_{recon} = \theta \underbrace{\left[1 - l_M(\tilde{p}) \cdot \prod_{j=1}^{M} cs_j(\tilde{p})\right]}_{\mathcal{L}_{\text{MS-SSIM}}} + (1-\theta).G_{\sigma_G^M}.\underbrace{\frac{1}{N}\sum_{p \in P} |X_i(p) - Y'_{recon}(p)|}_{\mathcal{L}_{\ell_1}} \tag{6}$$

where $Y'_{recon}$ is the reconstruction output. $l_M$ and $cs_j$ are derived from SSIM Equation 7 at scales $M$ and $j$.

$$\text{SSIM}(p) = \frac{2\mu_x\mu_y + C_1}{\mu_x^2 + \mu_y^2 + C_1} \cdot \frac{2\sigma_{xy} + C_2}{\sigma_x^2 + \sigma_y^2 + C_2} = l(p) \cdot cs(p) \tag{7}$$

$p$ stands for pixel. According to (Zhao et al., 2016), means and standard deviations are calculated using a Gaussian filter with a standard deviation $\sigma_G$, denoted $G_{\sigma_G}$. Moreover, MS-SSIM, a multi-scale form of SSIM, necessitates observing pixels at a neighbourhood of $p$ within a window with the same kernel size as $G_{\sigma_G}$, whereas calculating its derivative is infeasible in some boundary regions. Hence, the loss for the whole window is approximated by computing loss at its center pixel $\tilde{p}$.

## 2.3. Joint Objective

Eventually, the overall loss function $\mathcal{L}$ is formulated by:

$$\mathcal{L} = \mathcal{L}_{seg} + \lambda_0\mathcal{L}_{con} + \lambda_1\mathcal{L}_{MSCA} + \lambda_2\mathcal{L}_{MSP} + \lambda_3\mathcal{L}_{recon} \tag{8}$$

To determine the contribution of each loss to the overall loss value, we use $\lambda$ coefficients, which are set experimentally. The model is trained using Adam optimization with a learning rate of $1e-4$ and batch size 1 for 200 epochs. In addition, the code is developed in PyTorch, and the model is trained on a single RTX 3090 GPU.

## 3. Experimental Results

### 3.1. Dataset and Evaluation Metric

For performance evaluation, we use the BraTS 2018 dataset (Menze et al., 2014; Bakas et al., 2018), which is commonly used for benchmarking brain glioma segmentation pipelines with well-curated multi-institutional MRI data. This dataset includes 250 cases, each with four 3D MRI modalities (T1, T1c, T2, and FLAIR) rigidly aligned, resampled to $1 \times 1 \times 1$ mm isotropic resolution, and skull-stripped. The input volumetric spatial data size is $240 \times 240$ with 155 slices each. Annotations for the BraTS dataset include three tumor subregion classes: the enhancing tumor, the peritumoral edema, and the necrotic and non-enhancing tumor core. In our experiment, we follow the pre-processing steps and the same input size $(160 \times 192 \times 128)$ as suggested in (Azad et al., 2022c) to evaluate our network in terms of the Dice Similarity Coefficient (DSC).

### 3.2. Comparative Results

In Table 1, we provide the comparative results on each combination of the missing modality scenario. In comparison to the HeMIS(Havaei et al., 2016) and HVED (Dorent et al., 2019) strategies, our approach produces better segmentation results, specifically in the single modality (e.g., only T2) scenario. Moreover, SMU-Net (Azad et al., 2022c) proposes to decompose the representational space into style and content vectors and then strives to minimize the distribution difference between the full and missing modality path to perform knowledge distillation. Similarly, the ACN (Wang et al., 2021) takes the same perspective and performs the knowledge distillation in terms of both style and content modules but without separating them. Although both approaches strive to maximize the similarity between full and missing networks, they lack to include a mechanism to preserve the modality-specific features and distill such representation to the "missing modality" network. On the contrary, our approach proposes MSP tokens combined with the reconstruction head to guarantee such information preservation and performance gain. Quantitatively, compared to HeMIS and HVED, our approach significantly improves the performance by a 12-49 Dice score for single modality scenarios. In addition, compared to SMU-Net and the ACN approaches, the DSC score slightly rises by 0.5-2 Dice score. Visual segmentation results are presented in Figure 2 for single modality scenarios. It can be observed that the model can predict the brain tumour regions, whereas such prediction was not feasible by only relying on the input data without reconstructing the missing information.

### 3.3. Ablation Study

To analyze the effect of each module on the overall performance, we also provide an ablation study in Table 2. It can be observed that by eliminating each module from the proposed structure, the overall performance slightly decreases. The largest performance drop happens

Table 1: Performance comparison on the BraTS 2018 dataset using Dice metric. HeM, HVE, and MMC indicate the HeMIS, HVED, and our proposed MMCFormer models, respectively.

| Flair | T1 | T1c | T2 | HeM | HVE | ACN | SMU | MMC | HeM | HVE | ACN | SMU | MMC | HeM | HVE | ACN | SMU | MMC |
|---|---|---|---|---|---|---|---|---|---|---|---|---|---|---|---|---|---|---|
| | | Modalities | | | | Complete | | | | | Core | | | | | Enhancing | | |
| ○ | ○ | ○ | ● | 79.2 | 80.9 | 85.4 | **85.7** | 84.1 | 50.5 | 54.1 | 66.8 | 67.2 | **69.7** | 23.3 | 30.8 | 41.7 | 43.1 | **50.7** |
| ○ | ○ | ● | ○ | 58.5 | 62.4 | 79.8 | 80.3 | **80.4** | 58.5 | 66.7 | 83.3 | 84.1 | **86.6** | 60.8 | 65.5 | 78.0 | 78.3 | **79.0** |
| ○ | ● | ○ | ○ | 54.3 | 52.4 | 78.7 | **78.6** | 78.6 | 37.9 | 37.2 | **70.9** | 69.5 | 69.3 | 12.4 | 13.7 | 41.8 | 42.8 | **42.9** |
| ● | ○ | ○ | ○ | 79.9 | 82.1 | 87.3 | **87.5** | 86.2 | 49.8 | 50.4 | 66.4 | **71.8** | 70.0 | 24.9 | 24.8 | 42.2 | 46.1 | **49.6** |
| ○ | ○ | ● | ● | 81.0 | 82.7 | 84.9 | 86.1 | **86.3** | 69.1 | 73.7 | 83.2 | 85.0 | **86.7** | 68.6 | 70.2 | 74.9 | 75.7 | **79.1** |
| ○ | ● | ● | ○ | 63.8 | 66.8 | 79.6 | **80.3** | 80.2 | 64.0 | 69.7 | 83.9 | 84.4 | **86.6** | 65.3 | 67.0 | 75.3 | 75.1 | **78.7** |
| ● | ● | ○ | ○ | 83.9 | 84.3 | 86.0 | 87.3 | **87.7** | 56.7 | 55.3 | 70.4 | 71.2 | **72.0** | 29.0 | 24.2 | 42.5 | 44.0 | **48.1** |
| ○ | ● | ○ | ● | 80.8 | 82.2 | 84.4 | 85.6 | **85.9** | 53.4 | 57.2 | 72.8 | 73.5 | **74.0** | 28.3 | 30.7 | 46.5 | **47.7** | 46.9 |
| ● | ○ | ○ | ● | 86.0 | 87.5 | 86.9 | 87.9 | **88.0** | 58.7 | 59.7 | 70.7 | 71.2 | **72.4** | 28.0 | 34.6 | 44.3 | 46.0 | **48.3** |
| ● | ○ | ● | ○ | 83.3 | 85.5 | 87.8 | 88.4 | **88.7** | 67.6 | 72.9 | 82.9 | 84.1 | **86.7** | 68.0 | 70.3 | 77.5 | 77.3 | **79.5** |
| ● | ● | ● | ○ | 85.1 | 86.2 | **88.4** | 88.2 | 88.3 | 70.7 | 74.2 | 83.3 | 84.2 | **86.6** | 69.9 | 71.1 | 75.1 | 76.2 | **78.1** |
| ● | ● | ○ | ● | 87.0 | 88.0 | 87.4 | 88.3 | **88.5** | 61.0 | 61.5 | 67.7 | **67.9** | 67.8 | 33.4 | 34.1 | 42.8 | 43.1 | **50.7** |
| ● | ○ | ● | ● | 87.0 | 88.6 | 87.2 | 88.2 | **88.9** | 72.2 | 75.6 | 82.9 | 82.5 | **86.6** | 69.7 | 71.2 | 73.8 | 75.4 | **79.1** |
| ○ | ● | ● | ● | 82.1 | 83.3 | 86.6 | 86.5 | **86.8** | 70.7 | 75.3 | 83.2 | 84.4 | **86.7** | 69.7 | 71.1 | 75.9 | 76.2 | **79.2** |
| ● | ● | ● | ● | 87.6 | 88.8 | 89.1 | 88.9 | **89.0** | 73.4 | 76.4 | 84.8 | 87.3 | **87.4** | 70.8 | 71.7 | 78.2 | 79.3 | **80.1** |
| | | Mean | | 78.6 | 80.1 | 85.3 | **85.9** | 85.9 | 59.7 | 64.0 | 76.8 | 77.9 | **79.2** | 48.1 | 50.1 | 60.70 | 61.8 | **64.7** |

**(a) Ground Truth**  **(a) T1**  **(b) T1c**  **(c) T2**  **(d) Flair**

Figure 2: Segmentation results of the MMCFormer in single modality setting using BraTS 2018 dataset. The WT, edema, and necrotic regions are visualized with blue, red, and green colors. The results indicate that our method generates predictions that exhibit a smooth profile within each modality.

when the MSP tokens are eliminated, hence, showing the importance of preserving MS features. In fact, our model utilized each of these modules to ensure that the predictions of the missing modality network are in line with the predictions made by the full modality network in different levels of the network. This is particularly important when the missing modality network is trained with missing information and must rely on the full modality network as a reference. For instance, the MCA module helps the missing modality network

Table 2: Contribution of each MMCFormer module on the overall performance. For all experiments, only the T2 modality was used as input for the missing modality network.

| $\mathcal{L}_{\text{MCA}}$ | $\mathcal{L}_{\text{MSP}}$ | $\mathcal{L}_{\text{recon}}$ | WT | CT | ET | Average |
|:---:|:---:|:---:|:---:|:---:|:---:|:---:|
| ✗ | ✓ | ✓ | 83.7 | 69.2 | 49.7 | 67.6 |
| ✓ | ✗ | ✓ | 82.1 | 66.6 | 47.9 | 65.6 |
| ✓ | ✓ | ✗ | 84.0 | 68.9 | 50.3 | 67.7 |
| ✓ | ✓ | ✓ | 84.1 | 69.7 | 50.7 | 68.2 |

to align its feature representation in each level of the network and learn discriminative features that exist on the full-modality path. This helps to improve the accuracy and reliability of the predictions made by the missing modality network and leads to more robust performance in real-world applications. Additionally, the $\mathcal{L}_{\text{Consistency}}$ is calculated using the full modality network's prediction, which acts as a "teacher" to guide the missing modality network. This results in the missing modality network learning to generate more accurate predictions and reduces the risk of overfitting to the training data. Further ablation studies are provided in the Appendix section.

## 4. Conclusion

To address missing modality correction in an end-to-end manner, we proposed our MM-CFormer model, which utilizes a co-training strategy to perform knowledge distillation from the full-modality network into a missing modality one. To preserve modality-specific features, we proposed MSP tokens in conjunction with the reconstruction head to distill more discriminative features to the "missing modality" network. We also included a context agreement module to refine the feature representation in each scale of the co-training strategy.

## 5. Acknowledgment

This work was funded by by the German Research Foundation (Deutsche Forschungsgemeinschaft, DFG) – project number 455548460.

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

## Appendix A. Details on Network Architecture

The MMCFormer framework employs a pure Transformer-based hierarchical U-Net structure that builds upon a 3D efficient Transformer. As shown in Figure 3(a), there are two paths incorporated in the proposed model, namely the full-modality and the missing-modality paths. Both paths use the same structure but with different input data. In each path, the input image $X \in R^{H \times W \times D \times C}$ is first processed by a patch embedding module, which creates overlapping patch tokens of size $4 \times 4 \times 4$. These tokens are then processed by the encoder, which consists of three stacked encoder blocks. Each block contains two successive 3D efficient Transformer layers and a patch merging layer that merges $2 \times 2 \times 2$ patch tokens to reduce the spatial dimension while doubling the channel dimension. This hierarchical representation enables the model to acquire an overview of multiple scales and enhance the feature representations. In the decoder, the patch-expanding block increases the number of tokens by a factor of 2. The output of each patch-expanding block is then fused with the features from the corresponding encoder layer using a linear layer to recover spatial and fine-grained information. The resulting features are subsequently passed into two 3D efficient Transformer layers. Finally, a linear projection layer generates the final segmentation map. It is also worthwhile to mention that the full-modality network and the missing-modality network in our architecture contain 8.57 and 8.47 million parameters, respectively.

For the extra full-modality path, we define four modality-specific tokens entered into the bottleneck. We then design a modality reconstruction task to ensure that each token learns specific modality information. One of the four tokens is then randomly selected and fed into the single 3D efficient Transformer layer along with the resultant feature map obtained

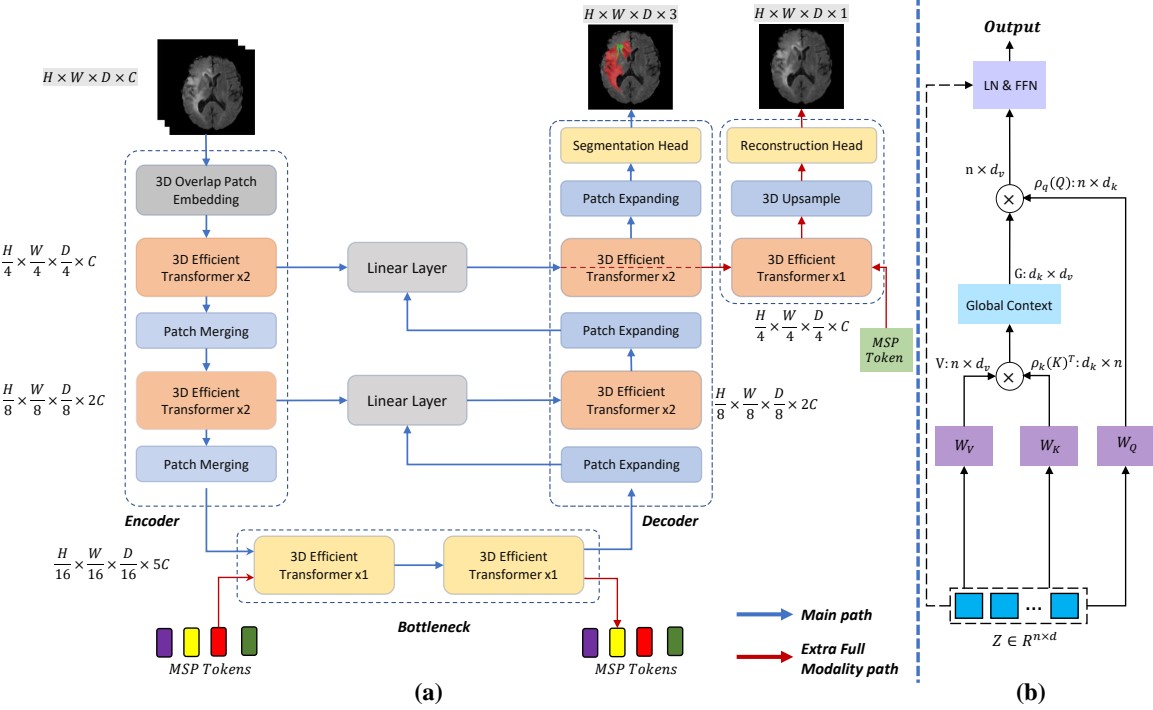

Figure 3: (a) The overview of the proposed network structure. (b) The structure of efficient Transformer.

from the last skip connection in order to reconstruct the specific modality assigned to the token. A 3D upsampling module followed by a linear reconstruction head is finally applied to acquire the modality-specific reconstruction image.

## Appendix B. Details of Network Training and Testing

In our proposed method, we use a two-path transformer network architecture to perform medical image segmentation, where the first network takes full modalities as input and the second network takes an arbitrary number of missing modalities as input. Similar to the literature work by Wang et al. (Wang et al., 2021) and Azad et al. (Azad et al., 2022c), we train the missing modality network for each combination of missing modalities and use its output to evaluate the final performance. This means that the missing modality network can be used to perform the segmentation task on an incomplete input (with missing modalities) at the test time. It worth mentioning that the input images for the two transformer networks come from the same subject, where the full modality path aims to help the missing modality network to learn how to recover the missing information that might occur during testing. To achieve this, during the training process, we inject images with the full set of modalities into the first network, and then manually discard some to serve as the input for the missing modality network. Once the models are trained, we only use the missing modality path to predict the segmentation map for new samples.

## Appendix C. The Importance of Global Context

The global context (GC) agreement utilized in our network imposes additional consistency on the network to ensure feature distillation hierarchically. In our experiments (subsection 3.3), we illustrated the contribution of this module to the overall performance of the network. Here we aim to visualize the global context matrix (attention map) of each path to highlight the similarity between these two paths by including or dropping the MSCA module. Figure 4 shows the GC of the full and missing modality path without imposing the MSCA module. It can be observed that there is a large variation between the attention map on both paths. In another word, this visualization shows that each network uses different feature representations to predict brain tumour segmentation maps. To enhance the feature representation by the missing modality path, we next include the MSCA module, results are shown in Figure 5. It is evident that the missing modality network now uses the full-modality path to recalibrate its feature representation to highlight the more informative features and enhance the importance of discriminative features that may exist in the network representation.

It is also worthwhile to mention that in our strategy, we include the MSCA module on each block of the encoder to ensure feature consistency in a multi-scale manner with a little overhead in terms of loss calculation. This strategy helps the network to learn the complex representations of the data in multiple stages, instead of a single stage. This is important because as the features get more abstract, the relationship between modalities also becomes more complex, and it is more difficult for the network to model the inter-modality relationship. By using hierarchical distillation, we allow the network to learn the relationship in multiple stages and at different levels of abstraction, which can lead to a better understanding of the data and more accurate predictions. Additionally, we believe that the added complexity (backward on the loss function) of using hierarchical distillation in every stage of the encoder is not considerable and is outweighed by the benefits it provides.

## Appendix D. Extreme Missing Modality

To further provide an insight into the effectiveness of our suggested framework for reconstructing the missing information, we offer quantitative and qualitative results in extremely missing modalities setting as presented in (Azad et al., 2022c). In this respect, we follow (Azad et al., 2022c) and design a scenario where during the training process the missing modality path only uses a single modality to learn the brain tumour segmentation map while benefiting from the full-modality path. During the test time, we only use the trained missing-modality path with a single modality as an input. Table 3 illustrates the quantitative results. For the comparison, we also include our baseline model, which consists of the co-trained Transformer-based U-Net model without any knowledge distillation mechanism. Quantitative results demonstrate that the MMCFormer surpasses the U-Net baseline method significantly. Besides that, our method achieves competitive results on all single modalities compared to the SOTA approaches. More specifically, in the T1ce and T2 modalities, our network surpasses the ACN and SMU-Net models by a large margin. Similarly, for the Flair modality, our method performs better compared to the ACN and HeMIS, HVED, and SMU-Net methods. In the T1 modality, we observed a large improvement compared to our baseline and approach but slightly less performance comparing to the ACN network.

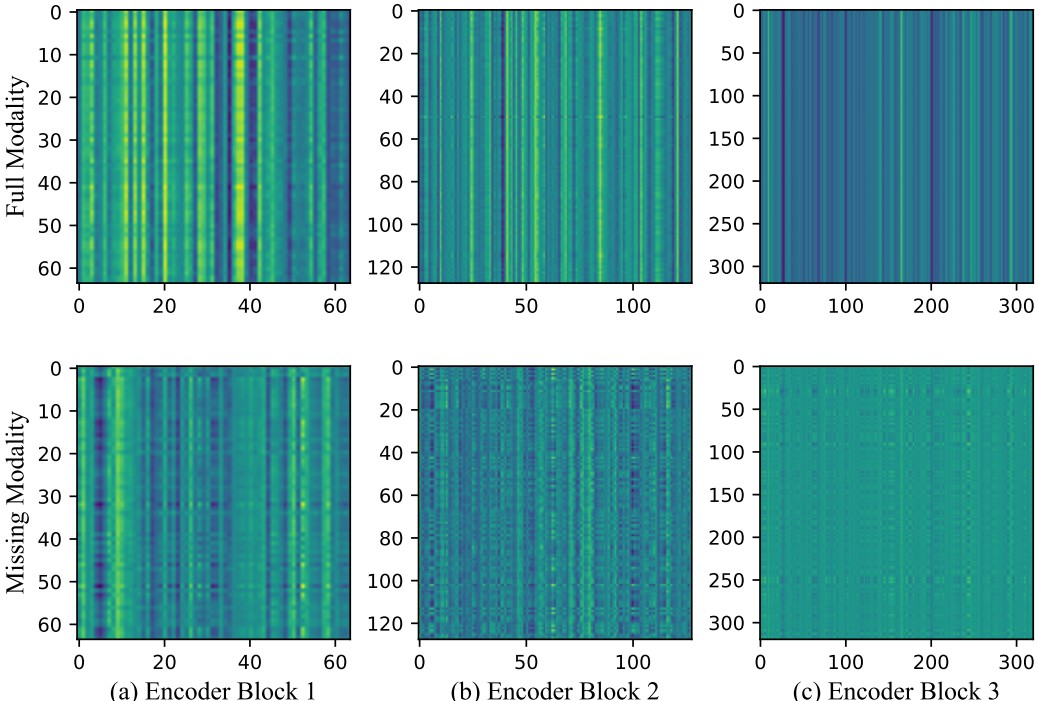

Figure 4:   Global context (GC) visualization without the proposed MSCA module.

The obtained results prove the efficacy of the suggested framework in learning task-specific patterns and recovering the missing information.

In order to validate the efficacy of incorporating a knowledge distillation strategy for enhancing performance and retrieving missing information, we conducted a comparative evaluation between our proposed method and the widely-recognized nnU-Net approach, as described by Isensee et al. (2019) (Isensee et al., 2019). Our results are presented in Table 3, which clearly demonstrates that, in comparison to conventional single-modality based methods, the knowledge distillation strategy can significantly enhance performance.

To visually analyze the qualitative performance of our suggested framework, we have provided the Grad-CAM (Selvaraju et al., 2017) attention maps in Figure 6-Figure 9 for each modality. In this respect, we trained and evaluated the network with an extremely missing modality setting. It can be seen that the MMCFormer localizes the brain tumour region with high confidence in a single modality case. This illustrates the effect of our co-training strategy for recovering the missing information. Additionally, the T1 modality is more effective for characterizing the structural information and less informative for detecting for core tumour region, however, our MMCFormer with a T1 modality can recognize the tumour region with high confidence which indicates the importance of our suggested modules for recovering the missing information. Similarly for other modalities, we can observe that the activation map has a high magnitude around the tumour region and it highly overlaps with the grand truth annotation mask. On top of that, we can observe that the MMCFormer attention has less variance and it is in line with the ground truth map, which shows the effect of reconstructed features for precise boundary prediction.

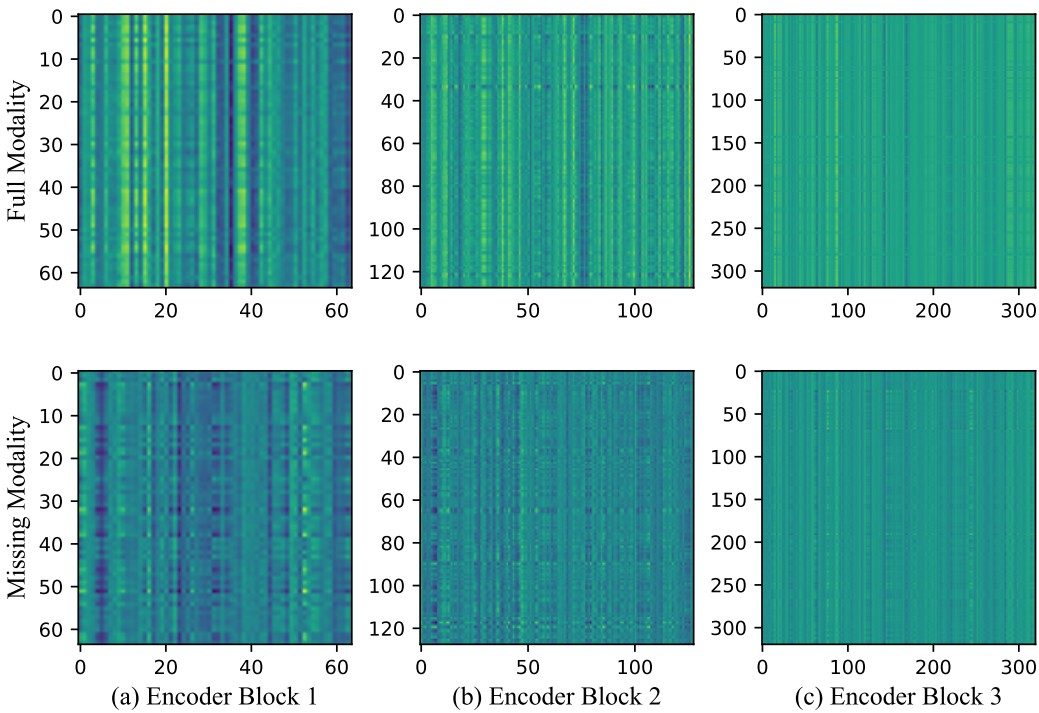

(a) Encoder Block 1    (b) Encoder Block 2    (c) Encoder Block 3

Figure 5:   Global context (GC) visualization with the proposed MSCA module.

Table 3: Quantitative results of the proposed MMCFormer model against the SOTA approaches on BraTS 2018 series with extreme missing modality scenario. We use the average of the whole, enhanced and core tumour segmentation scores to report the dice score for each modality.

| Article | Dice score | | | | |
|---|---|---|---|---|---|
| | T1 | T1c | T2 | FLAIR | AVG |
| Our Baseline | 61.2 | 77.3 | 60.1 | 58.2 | 64.2 |
| U-HeMIS (Havaei et al., 2016) | 16.7 | 59.2 | 36.0 | 51.5 | 48.8 |
| HVED (Dorent et al., 2019) | 34.4 | 64.8 | 55.2 | 52.4 | 51.7 |
| ACN (Wang et al., 2021) | **63.8** | 80.3 | 64.6 | 65.3 | 68.5 |
| SMU-net (Azad et al., 2022c) | 63.3 | 80.9 | 65.3 | 68.4 | 69.4 |
| nnU-Net(Isensee et al., 2019) | 62.4 | 77.6 | 61.3 | 61.8 | 65.7 |
| MMCFormer | 63.6 | **82.0** | **68.2** | **68.6** | **70.6** |

## Appendix E.  Reconstruction Head

In our strategy, we included the reconstruction head to reconstruct each MRI modality during the training time. The objective of the reconstruction head in our design is to guide the network through the learning modality-specific feature and transfer such discriminative information to the missing modality network. To ensure the performance of our reconstruction head as an auxiliary task, we visualize the reconstructed modality in Figure 10 along

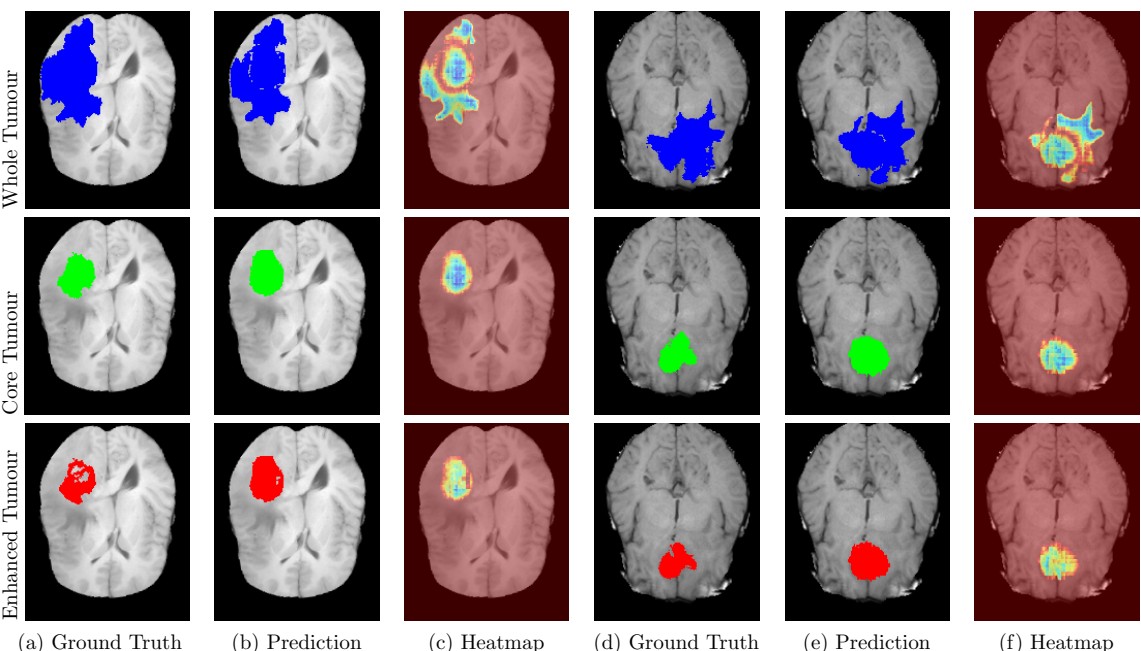

(a) Ground Truth  (b) Prediction  (c) Heatmap  (d) Ground Truth  (e) Prediction  (f) Heatmap

Figure 6: Visualization of the activation map on T1 modality using our suggested MMC-Former.

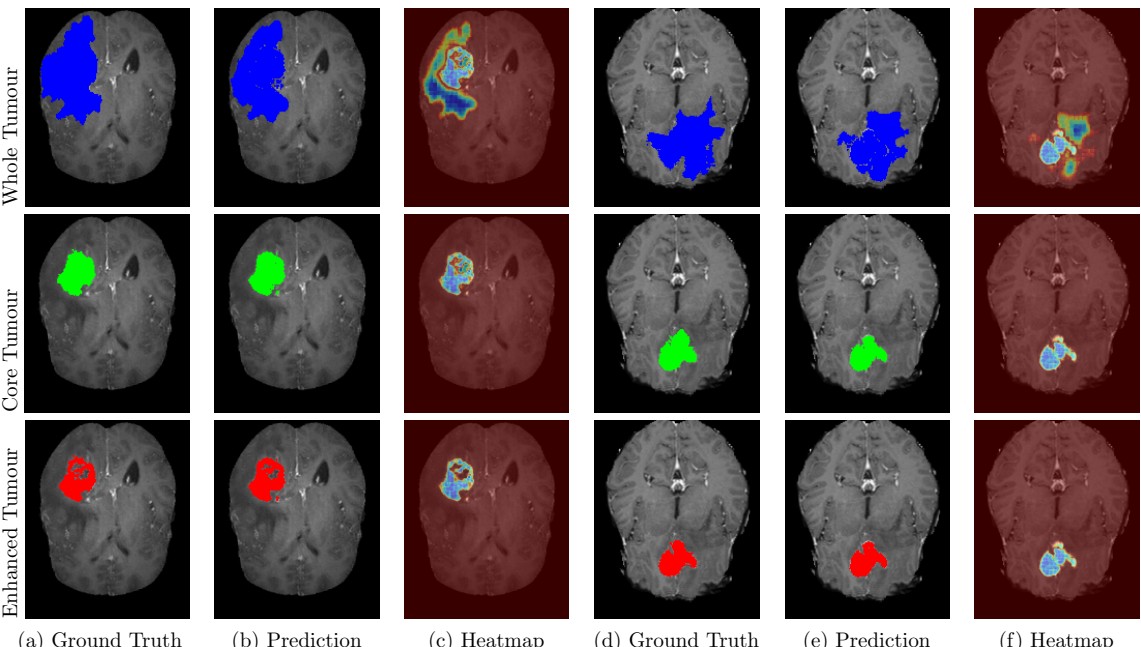

(a) Ground Truth  (b) Prediction  (c) Heatmap  (d) Ground Truth  (e) Prediction  (f) Heatmap

Figure 7: Visualization of the activation map on T1c modality using our suggested MMC-Former.

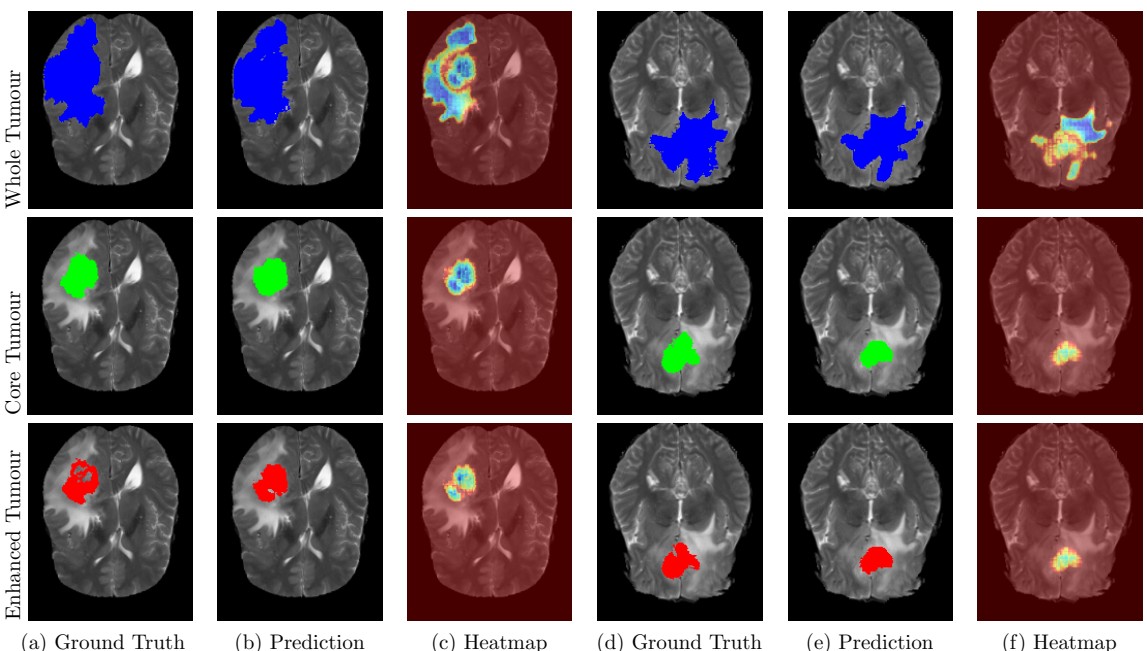

Figure 8: Visualization of the activation map on T2 modality using our suggested MMC-Former.

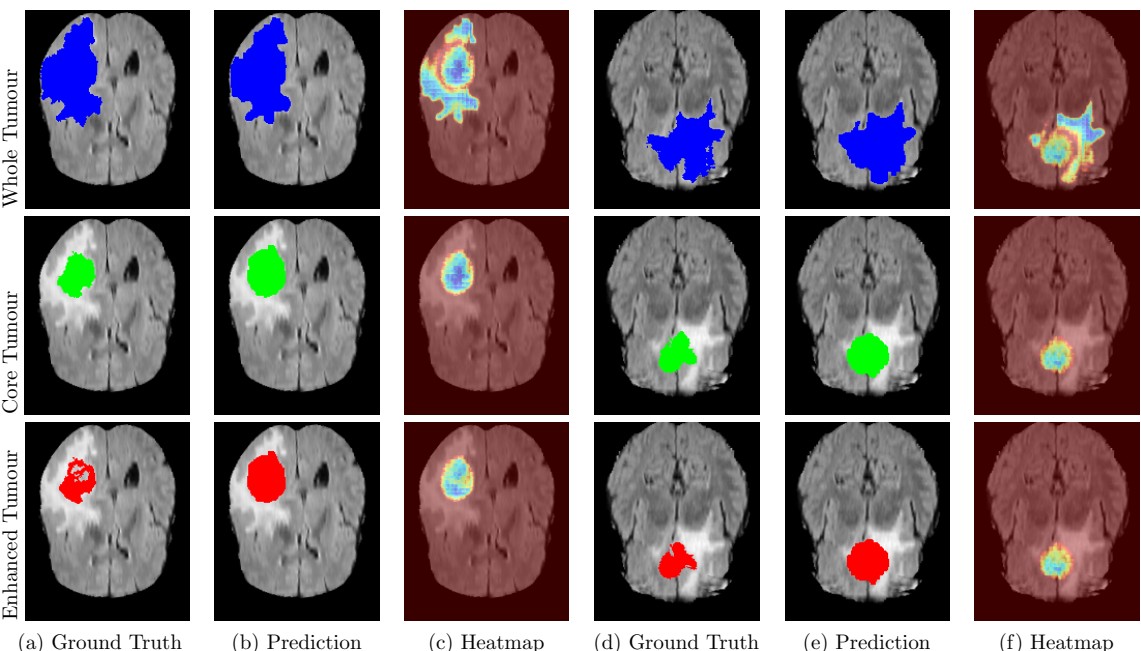

Figure 9: Visualization of the activation map on Flair modality using our suggested MM-CFormer.

with the error map. We use a pixel-wise difference for calculating the error mask. It can be seen that for all modalities, our network manages to reconstruct the images with good quality and an acceptable error rate. There is also a high potential for more fine-grained reconstruction by increasing the complexity of the reconstruction head, however, the reconstruction is not the main task in our design and therefore we use a simple reconstruction head to fulfill the objective. It is also worthwhile to mention that in our strategy we utilize the combination of MS-SSIM and L1 distances for the reconstruction loss. Although including the MS-SSIM loss enhances the reconstruction loss, it requires more computation complexity. Hence, one might consider such complexity in their use case.

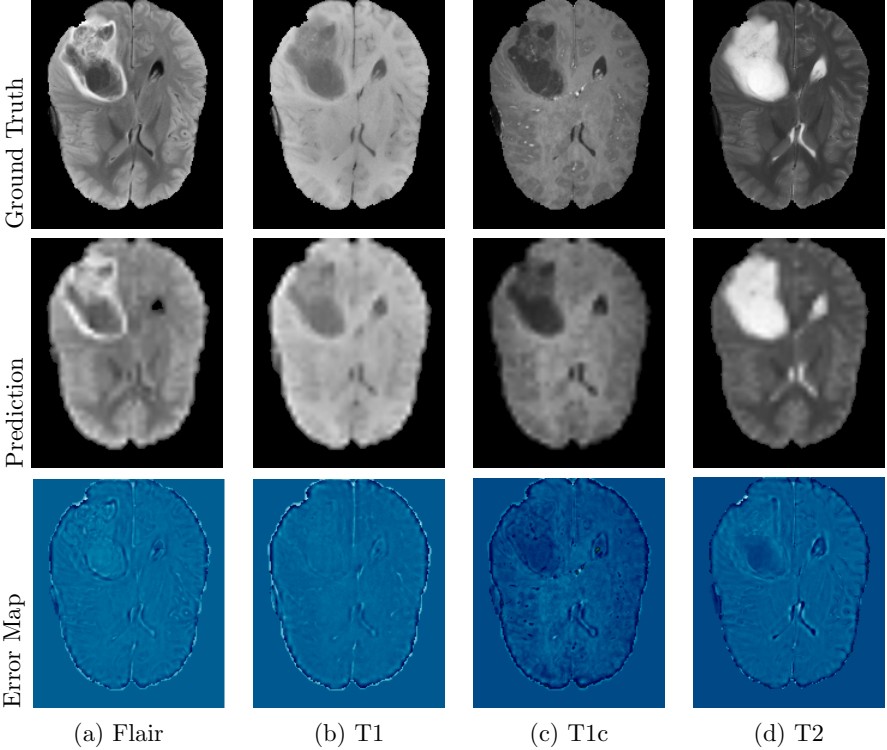

(a) Flair      (b) T1      (c) T1c      (d) T2

Figure 10: Visual reconstruction of Flair, T1, T1c, and T2 modalities, along with the error map between the reconstructed results and the ground truth.

