# OpenReview forum: "MMCFormer: Missing Modality Compensation Transformer for Brain Tumor Segmentation"
_MIDL.io/2023/Conference — MIDL 2023 Oral_

### Official Review · Reviewer_2Y5x · 2023-02-02

**Confidence:** 4
**Preliminary Rating:** 4
**Recommendation:** Poster

**Summary:**

As authors point out 4 modalities (T1, T1c, Flair,T2) bring the highest segmentation performance when used as input to a segmentation network. However, the 4 modalities ar epresent in datasets like Brats but not all 4 are always present. To address the missing modality problem, the authors propose a transformer-based architecture using knwoledge distillation training principle. Two network branches are trained at the same time: a branch that takes all 4 modalities, and a branch that takes less than4 input modalities. Both branches predict a segmentation map which compared with the ground truth.

Several losses are introduced to match the latent representations learned by the two branches. Context loss that matches branches predictions, two new losses matching attention elements and tokens, and auxilary reconstruction loss.

The method is tested on Brats datset.

**Strengths:**

The method addresses the clinically relevant problem of semantic segmentation under missing MRI modalities settings.
Transformer-architecture principles are used to construct a segmentation network.
New attention query, key, and token losses are introduced to foster knowledge distillation.
Overall, the proposed design improves the accuracy compared to SOTA methods.
Ablation is performed for the proposed losses.


**Weaknesses:**

I have only one high-level comment. It would be useful to know what the numbers in Tab. 1 are for a SOTA in semantic segmentation for Brats (nn-Unet), in order to understand better how big of a problem the missing modalities are. It seems that using two (out of four) modalities T1c and Flair leads to pretty competitive results. So I wonder whether nnUnet with these two modalities provides accuracy better than all of your models? I can believe though that your method (and other missing modality methods) can be better than nnUnet which has only a single modality as an input.



**Deanonymize Review:**

no

**Paper Type:**

methodological development

**Questions To Address In The Rebuttal:**

Please address all the point above.

Minor stuff:

In captions to figure 2, the authors say "The WT, edema, and necrotic regions are visualized with blue, red, and green colors."
There is a mistake. You have edema in blue, necrotic core in green, and enhancing tumor in red.

I suggest authors remove mathematical notations in this sentence "Xf = {∀Xi, i ∈ M}, and the missing modality as Xm = {∀Xi, i ∈ M′}".
Strictly speaking, this is not correct. Alternatively, if you want to keep the math, you can define it via set-builder or union notations.

Two typos in Eq 4, C_m instead of C_2, and missing f index in the denominator

Is there MSCA loss on the decoder branches?

---

### Official Review · Reviewer_1npZ · 2023-02-03

**Confidence:** 5
**Preliminary Rating:** 4
**Recommendation:** Oral

**Summary:**

The key contribution is an approach to handle missing modalities for segmentation and applied to the public BRATs dataset. The paper is fairly well written and comparison experiments with similar methods that address heteromodality segmentation are performed. The method introduces a co-training approach for handling missing modalities by training a pair of transformers, one with full and the second with missing modalities and performs hierarchical token distillation at multiple scales to match the features. The results indicate accuracy improvements particularly as the number of available modalities is increased compared to other benchmarks.

**Strengths:**

+ Fairly well written paper
+ Addresses a need in medical image analysis with missing modalities with a transformer based approach to better utilize the anatomic context in the modeling.
+ Comparison benchmarks are used to demonstrate improved method performance

**Weaknesses:**

+ Ablation analysis is limited and needs to be more comprehensive. In particular, why do you need hierarchical distillation, why not just a few stages - using hierarchical distillation in every stage adds complexity in network training and resources. What is the gain to be had?
+ Choice of bottleneck layer for matching modality features from the two networks does not make a lot of sense, especially since the networks process different modalities. Why not task relevant features that are closer to output
+ Similarly ablation experiments don't analyze the utility of the consistency loss.
+ Details of network training and testing could have been more detailed to show how you trained with only 250 cases and a more complex network compared to the others. How many were set aside for training and testing.

**Deanonymize Review:**

no

**Detailed Comments:**

Please see above for details

**Paper Type:**

methodological development

**Questions To Address In The Rebuttal:**

It's recognized that the detailed ablation study is hard to do for rebuttal. However, at least adding consistency loss in the analysis and providing better rationale for the choice of bottleneck layer for modality specific loss is needed.
Also please provide more details of training and testing and details of network size compared to the other networks to understand generalization ability in relation to the network capacity.

---

### Official Review · Reviewer_zJwz · 2023-02-04

**Confidence:** 3
**Preliminary Rating:** 3

**Summary:**

This paper addresses the issue of missing modality in medical image segmentation. In the proposed method, two networks, consisting of identical efficient transformers, are trained jointly. The first network is fed with images with full-modalities, while the second with incomplete ones. Three regularization losses are proposed, including a correlation based alignment loss, and  modality specific loss, and a reconstruction loss. Experiments show improved results compare to several baselines.

**Strengths:**

The paper addresses the issue of missing modality in medical image segmentation which is important but largely ignored by the research community. It proposes a method using two identical networks to learn the information from complete and missing data via a co-training approach. For their transformer-based segmentation network, the authors employ several strategies to improve performance: an efficient self-attention mechanism to reduce computational overhead, two separate networks to learn complete and missing information, two losses to enhance information flow between networks, as well as an auxiliary reconstruction loss to preserve modality specific information.

This paper is well-written and easy to follow. It includes a brief diagram of the proposed network as well as a more detailed one in the Appendix. The mathematical formulation is clear and makes sense.

The authors confirm the effectiveness of the proposed method on BraTS 2018 dataset, and results seem to support their claims. In the Appendix, results with extremely missing modalities are also provided.

**Weaknesses:**

For the methodology, if I understand correctly, the input images for two transformer networks are from the same subject. That is saying the authors inject images with full-set modalities into the first model, while manually discard some as the input of the second network. This makes sense in some particular cases, but would limit this method to apply in real cases where one only has subjects with only different missing modalities and few of them have a complete modality set.

The paper lacks details about the experimental setup. Comparing different methods for medical image segmentation is not trivial, and providing details on the chosen methods would help ensure the reproducibility and the fairness of the comparison.

The authors only report results in one dataset, which might be insufficient to validate their approach.

**Deanonymize Review:**

no

**Detailed Comments:**

* The paper lacks a proper discussion of main results and limitations of the work.

* Eq. (5)-(7) do not really provide useful information (these are well known concepts in linear algebra). In my opinion, it would have been preferable to use this space for some of the results added to the Appendix.

**Paper Type:**

methodological development

**Questions To Address In The Rebuttal:**

* Does the second network take input images with arbitrary number of missing modalities? Is their a predefined ordering for these modalities? How are the modalities organized in the input?

* It seems the two network has a consistency loss on the decoder part, which means the input of the two networks should refer to the same subject (image). Is it a way to train this method with different subjects as input of the two networks?

* Eq. 7 describes the global context of the two networks. Would it be possible to visualize this?

* Can the proposed method be applied to a CNN based network, if so, which modifications should be considered? If not, please explain why.

---

### Meta-Review · Area_Chair_n4YE · 2023-02-23

**Recommendation:** Accept (Poster)
**Confidence:** 5

**Metareview:**

The reviewers in general agree that this paper addresses a practically important problem (missing modalities) for semantic segmentation. The reviewers also are in agreement that this paper is in general well written. While the results are limited to the BRATS dataset, they were deemed to be sufficient to provide a proof-of-concept.

---

### Meta-Review · Program_Chairs · 2023-02-28

**Recommendation:** Accept (Oral)
**Confidence:** 4

**Metareview:**

This a great well written study, with interest to the community. I recommend it for an oral presentation.